# Pinacolone-Alcohol Gas-Phase Solvation Balances as Experimental Dispersion Benchmarks

**DOI:** 10.3390/molecules25215095

**Published:** 2020-11-03

**Authors:** Charlotte Zimmermann, Taija L. Fischer, Martin A. Suhm

**Affiliations:** Institut für Physikalische Chemie, Georg-August-Universität Göttingen, Tammannstr. 6, 37077 Göttingen, Germany; czimmer2@gwdg.de (C.Z.); tfische1@gwdg.de (T.L.F.)

**Keywords:** dispersion, ketone–alcohol complexes, density functional theory, hydrogen bonds, molecular recognition, vibrational spectroscopy, gas phase, benchmark, pinacolone

## Abstract

The influence of distant London dispersion forces on the docking preference of alcohols of different size between the two lone electron pairs of the carbonyl group in pinacolone was explored by infrared spectroscopy of the OH stretching fundamental in supersonic jet expansions of 1:1 solvate complexes. Experimentally, no pronounced tendency of the alcohol to switch from the methyl to the bulkier *tert*-butyl side with increasing size was found. In all cases, methyl docking dominates by at least a factor of two, whereas DFT-optimized structures suggest a very close balance for the larger alcohols, once corrected by CCSD(T) relative electronic energies. Together with inconsistencies when switching from a C4 to a C5 alcohol, this points at deficiencies of the investigated B3LYP and in particular TPSS functionals even after dispersion correction, which cannot be blamed on zero point energy effects. The search for density functionals which describe the harmonic frequency shift, the structural change and the energy difference between the docking isomers of larger alcohols to unsymmetric ketones in a satisfactory way is open.

## 1. Introduction

In nature, directional hydrogen bonds to carbonyl groups [1,2] are frequent, for instance in proteins, DNA or other biopolymers [3,4]. London dispersion interactions are less directional, but at least as omnipresent [5]. An accurate and detailed theoretical description of these interactions and their cooperation or competition is urgently needed. As in any complex interplay, there is a risk of error cancellation. One may easily get the right answer for the wrong reason. The situation calls for systematic isolation attempts with respect to the different contributions. This can be achieved by the study of a series of small hydrogen-bonded complexes at low temperature in the supersonically expanded gas phase by rotational and vibrational spectroscopy [6,7,8]. Even at low temperature, anharmonic zero point vibrational energy (ZPVE) still complicates the comparison between electronic structure theory and experimental information on the relative energy of different molecular arrangements [9]. A more direct test of the potential energy landscape would be very desirable.

This has led to the concept of ketone solvation balances, which were introduced for acetophenone and its derivatives in combination with alcohols as hydrogen bond donors [10,11] and tested for other ketones [12,13]. The idea is to have two very comparable lone electron pairs available at the acetophenone oxygen, to which alcohols can either dock from the phenyl or from the alkyl side, with little difference in ZPVE. Besides the intrinsic preference of a docking alcohol for the methyl side due to the more favorable local hydrogen bond geometry [11], the alkyl group of the alcohol will interact dispersively (and by Pauli repulsion) with the two ketone substituents and thus contribute to the preference for one of the docking sides. These secondary interactions through space are able to tip the balance towards phenyl side docking [11]. The comparison of different alcohols and acetophenones thus provides information on London dispersion interactions competing with the electronic and zero-point vibrational local hydrogen bond effects, which still largely govern the position of the alcoholic OH vibration. The latter is used to spectrally discriminate the docking isomers and it also contains further information on the competition of forces, because hydrogen bonds can be distorted by distant interactions of the donor molecule. Experimental information on the docking preference comes from the relative abundance of the docking isomers in the quasi-equilibrium established by cooling collisions in a supersonic jet expansion, down to some conformational freezing temperature $T_{c}$ (roughly 30 to 150 K, depending on low (1 to 5 kJ${\mathrm{mol}}^{-1}$) and narrow interconversion barriers [10,14,15]) and can thus only be predicted with a large tolerance.

The results of such studies can be used to benchmark the ability of different density functionals to predict the interplay of hydrogen bonding with distant London dispersion and Pauli repulsion, by simply comparing the predictions to experiment. This can be done strictly at the level of observables, without consulting any energy decomposition models [16,17,18], although the latter are helpful in the interpretation of the findings. A functional which gives the right answer for the right reason in the popular harmonic approximation for vibrations must be able to predict the splitting of the OH stretching vibrations between the docking isomers (because anharmonic effects by construction largely cancel when comparing the isomers) and the relative abundance of the isomers with a reasonable conformational temperature. As a third test, high level single point wavefunction calculations (for which Hessian calculations to reproduce the spectrum would be too costly) at the optimized DFT minima should confirm the energy predictions in a qualitative sense. If at least one of these three diagnostics fails, the DFT functional performance can be proven to be poor down to a sub-kJ/mol accuracy threshold. This was the case for one out of six pairings of aromatic ketones with alcohols in the first systematic study [11], for the otherwise most successful B3LYP-D3 functional. By using the second-most stable and less compact predicted structure in this particular case, the performance could actually be rescued [11]. This former systematic investigation thus suggests a mildly erroneous preference of the B3LYP-D3 functional (at least for a standard def2-TZVP basis set), and to a lesser extent also the TPSS-D3 approach, for compact structures. Other explored functionals such as M06-2X failed the aromatic ketone balance test in several aspects [11] and need not be considered further.

The hypothesis that B3LYP-D3 and TPSS-D3 show an (almost) acceptable performance for ketone dispersion balances obviously calls for further falsification attempts and this is the task of the present study which involves the purely aliphatic pinacolone (see Figure 1), where the phenyl group in acetophenone is replaced by a *tert*-butyl (*t*Bu) group. This removes aromatic–aliphatic dispersive interactions and brings in more bulky donor-acceptor constellations. Cyclopentanol (CpOH) is introduced as a further, more disk-like and flexible alcohol, in addition to methanol (MeOH) and *tert*-butyl alcohol (*t*BuOH), which have been previously explored with acetophenone [11]. Pinacolone monomer does not have a plane of symmetry [19], but in combination with low planarization barriers (Appendix A) and symmetry-breaking alcohol coordination, this should not lead to additional complications in the analysis. Indeed, there are significant variations of the hydrogen bond angle α and the dihedral angle τ (see Figure 1) with alcohol substitution. These promise to explore the hydrogen bonding potential of carbonyl groups far away from the intrinsic in plane preference.

In this work, we show that, in alcohol–pinacolone balances, the methyl docking side is consistently preferred. According to exploratory calculations, this may extend to many alcohols beyond the experimentally investigated ones. Further, we show that the predictive quality of the two density functionals which were successful for acetophenone (B3LYP and TPSS) decreases with the size of the alcohol, including significant failures for the largest (CpOH). The proposed assignments and observed trends are discussed and an analysis of dispersion interactions on the docking side preference is presented. We provide initial evidence that some of the superficially satisfactory DFT performance for ketone balances must be fortuitous.

## 2. Results and Discussion

We start with the theoretical description of alcohol–pinacolone 1:1 complexes at the level of DFT before comparing to the experimental findings and finally consulting wave-function theory.

### 2.1. Density Functional Predictions

From now on, the abbreviations Pin for the studied ketone pinacolone and MeOH (methanol), *t*BuOH (*tert*-butyl alcohol) and CpOH (cyclopentanol) for the solvating alcohols are consistently used. In Figure 1, two angles α and τ describing the hydrogen bond geometry are introduced. The hydrogen bond angle between the hydrogen bonded H and the carbonyl group has a local, sp${}^{2}$-explainable preference for ≈120${}^{\circ }$. The dihedral angle τ describes the out of plane twist of the docking alcohol OH with respect to the carbonyl plane, with two local preferences near 0${}^{\circ }$ and 180${}^{\circ }$. Any deviations from these local preferences due to global interactions sensitively affect the hydrogen-bonded OH stretching wavenumber.

As detailed in Appendix A, all six experimentally investigated 1:1 complexes show a narrow distribution for α (115–124${}^{\circ }$) at the four investigated DFT levels (D3-corrected B3LYP and TPSS with triple and quadruple zeta basis sets). τ deviates from planarity with increasing size of the alcohol, in steps of roughly 10${}^{\circ }$ from MeOH over *t*BuOH to CpOH. On the *t*Bu docking side of Pin, even MeOH is already displaced by 35–37${}^{\circ }$, due to the bulkiness of the substituent, whereas the Me docking displacement is less than 10${}^{\circ }$ for MeOH.

The structural trends are reflected in the calculated OH stretching wavenumbers (see Appendix A), which are consistently lower for Me docking for all three alcohols, whereas the trend with increasing alcohol size is comparatively weak, relative to the overall hydrogen bond shift. This assists a straightforward interpretation of the experimental spectra.

The energy differences between Me and *t*Bu docking sides fall between 0 and 3 kJ${\mathrm{mol}}^{-1}$, always preferring the Me side, as shown in Figure 2. The narrow corridor of ±0.2
kJ${\mathrm{mol}}^{-1}$ in the figure (gray lines) illustrates that it makes almost no difference whether harmonic ZPVE is included or not. The effect of basis size extension is similarly small. This is very favorable for a direct judgement of the DFT functional in terms of the predicted electronic energy difference without worrying about major (anharmonic) zero point energy or basis set effects which can both be quite significant when looking at absolute energies and frequencies [10,20].

The predicted spread in docking energy difference of about 2.5
kJ${\mathrm{mol}}^{-1}$ across the systems promises a large variation of the experimental abundances, but the absence of a sign reversal (corresponding to an absence of data points in the upper right quadrant of Figure 2) despite varying the alcohol size from 1 to 5 carbon atoms is surprising. An explorative search for almost 20 other alcoholic donors (see Appendix A) confirms this systematic bias. The steric disadvantage of the *t*Bu side of Pin together with the flexibility of alcohols provides possible explanations. The latter allows the alcohol to dock on the sterically more accessible Me side and at the same time to exploit London dispersion interaction with the *t*Bu side. A good example is benzyl alcohol, where the Me sided structure is almost 2 kJ${\mathrm{mol}}^{-1}$ more stable, because the benzyl group can still interact favorably with the *t*Bu group of the Pin while the OH group is docking to the Me side of Pin.

Another important feature of carbonyl balances is the feasibility of the isomerization under supersonic jet expansion conditions. A transition state search between the two competing structures for MeOH–Pin yielded an interconversion barrier height of about 3 kJ${\mathrm{mol}}^{-1}$ when viewed from the *t*Bu docking structure. The interconversion path is distinctly out-of-plane, relaxing the hydrogen bond angle α while switching between small and large τ. This is similar to previous findings for acetophenone [11] and its derivatives and supports a feasible interconversion under supersonic jet conditions, with $T_{c}$ values significantly below the starting temperature of the expansion. However, the more numerous the contacts between the residue and Pin are, the larger this barrier may become. This is one reason this work focuses on small alcohols to establish the performance of the DFT functionals.

Before switching to experiment, two important observable predictions need to be explored. One is a sufficiently robust infrared cross section ratio for the docking isomers, which is a precondition for reliable experimental abundance determinations from spectral intensities. As shown in Appendix A, the basis set and functional dependences are modest and the trends are smooth, such that this variation and the double-harmonic approximation are not expected to be critical for the theory-experiment comparison.

The most important theoretical assignment aid concerns the predicted positions and differences or splittings of the OH stretching fundamental vibrations. While the harmonic approximation is too crude for absolute predictions, the harmonic Me-*t*Bu differences involve systematic cancellation of anharmonic contributions for similar docking environments. Furthermore, the structural effect of increasing alcohol size is qualitatively similar on both docking sides, as pointed out above, and should translate into relatively uniform wavenumber splittings as a function of the number of alcoholic C atoms. This is illustrated in Figure 3. In all cases, the Me-docking wavenumber is lower, corresponding to a uniformly negative $\Delta \omega_{\mathrm{Me}-t\mathrm{Bu}}$ value and facilitating experimental assignment. The size of the splitting exceeds the spectral resolution and band width [11] by more than an order of magnitude, which is also favorable.

With a single exception (TPSS for CpOH for the larger basis set), all predicted harmonic splittings are within ± 12 ${\mathrm{cm}}^{-1}$ of the average value of −42 ${\mathrm{cm}}^{-1}$ and there is only a weakly decreasing trend for the splitting with increasing alcohol size. For CpOH, predictions range from −50 to −63 ${\mathrm{cm}}^{-1}$. These variations are also reflected in the τ angle (Appendix A). They are robust with respect to cross-over re-optimization and indicate a slight TPSS-bistability of the structure depending on the basis set. Anticipating the experimental (anharmonic) result reported in the next section (blue symbols and lines in Figure 3), the larger basis set result appears less likely in absolute numbers but more likely in terms of the trend. Even beyond this outlier, it is clear that the alcohol size trends are not predicted perfectly, thus underscoring the benchmarking potential of this study.

### 2.2. Experimental Results

In Figure 4, the experimental infrared spectra for helium supersonic jet expansions of Pin with MeOH (green), *t*BuOH (orange) and CpOH (blue) are shown. They feature the rovibrationally broadened alcohol monomer OH stretching bands (MeOH, *t*BuOH, CpOH), the downshifted hydrogen-bonded homodimer signals (${\left(\mathrm{MeOH}\right)}_{2}$, ${\left(t\mathrm{BuOH}\right)}_{2}$ and ${\left(\mathrm{CpOH}\right)}_{2}$), as well as the narrow bands of the mixed complexes with docking isomerism $O_{\mathrm{Me}}$ and $O_{t\mathrm{Bu}}$. For MeOH, $O_{\mathrm{Me}}$ and $O_{t\mathrm{Bu}}$ are spectrally downshifted compared to the respective homodimer band, as one might expect from an intrinsically stronger OH⋯O=C interaction, whereas, in the case of *t*BuOH and CpOH, they are upshifted. This is already an experimental sign for competition between hydrogen bonding and more global London dispersion interactions. Even when the alcohol is in Me docking position, where there is no sterical crowding, it is displaced out of the ketone plane to maximize interaction with the *t*Bu group (see Appendix A). When CpOH is combined with acetone, which lacks the *t*Bu group (see Appendix A), the homodimer and mixed dimer signals actually overlap. This is partly due to less competition from dispersion interaction with the other side of the ketone for the hydrogen bond.

The more downshifted mixed dimer band $O_{\mathrm{Me}}$ in Figure 4 is always significantly stronger and based on the robust DFT predictions for structure (Appendix A) and downshift (Figure 3 and Appendix A), it must be due to Me docking, as implied by the label. Given that its spectral visibility (Appendix A) is at best twice that of the *t*Bu isomer, it must also be the more stable isomer, in agreement with the DFT computations (Figure 2).

The experimental shift between $O_{\mathrm{Me}}$ and $O_{t\mathrm{Bu}}$ spans a relatively narrow range of 30 to 40 ${\mathrm{cm}}^{-1}$(Figure 3 and Appendix A), which roughly matches the DFT prediction window, except for the TPSS outlier. In many cases, the DFT splitting is somewhat larger than the experimental one, which matches the general overestimation of hydrogen bond shifts by most density functionals. The subtle alcohol substitution trend in the splittings (Figure 3) is not well reproduced, being monotonically decreasing for the DFT predictions and non-monotonic in the experiment, but, considering the superposition of Me and *t*Bu trends, this appears acceptable and does not complicate the spectral assignment.

In Figure 5, the experimentally determined downshifts from the monomer OH fundamental are plotted against the corresponding calculated ones. The fact that all correlation points stay below the diagonal line confirms the systematic overestimation of DFT downshifts, which is more pronounced for TPSS than for B3LYP [11] and only in part due to anharmonicity. The slope of the data points matches the diagonal for methanol (dashed arrows connect isomers), but it becomes flatter for the more bulky alcohols. This indicates that the DFT calculations overestimate the hydrogen bond weakening by bulkiness (dispersion and/or exchange repulsion). Note that non-isomeric acetone docking results [21] (for CpOH, see Appendix A) included in the figure also fit the Pin data for Me docking.

The CpOH–Pin case is also suspicious in terms of the B3LYP energy gap between Me and *t*Bu docking. Based on Figure 4, Me docking should be substantially more stable, even more so if statistically formed conformations freeze rather early in the expansion. However, the predicted energy difference is ≤ 0.5
kJ${\mathrm{mol}}^{-1}$ (see Appendix A), far too low for such an imbalance. Attempts to rescue the situation in analogy to the acetophenone balance study [11] by searching for metastable minima on the DFT hypersurfaces failed (see Appendix A for details). In this context the pseudorotational isomerism of the axial CpOH monomer should be briefly discussed. There are two nearly isoenergetic (B3LYP-D3(BJ,abc)/def2-TZVP energy difference less than 0.1
kJ${\mathrm{mol}}^{-1}$) and isospectral isomers (about 4${\mathrm{cm}}^{-1}$
wavenumber difference, leading to slight shift uncertainty), depending on the position of the axial OH in the envelope conformation. This has caused uncertainty in rotational [22] and has also been addressed in liquid state spectroscopy [23]. However, structure optimizations indicate that this subtle isomerism is relaxed in the complexes with Pin, leading to uniform axial gauche results (or energetically > 2 kJ${\mathrm{mol}}^{-1}$ higher conformations). The problem is thus more fundamental, as the following analysis supports.

For this purpose, the experimental abundance is compared to the predicted B3LYP energy difference for all three investigated systems by calculating a concentration ratio $c_{\mathrm{Me}}/c_{t\mathrm{Bu}}$, which follows from the experimental intensity ratio and the def2-TZVP absorption cross sections (see Table 1). The maximum and minimum values for $I_{\mathrm{Me}}/I_{t\mathrm{Bu}}$ from a Monte Carlo integration program [24] generate a maximum and minimum value for $c_{\mathrm{Me}}/c_{t\mathrm{Bu}}$ which is further transformed to a (semi-)experimental $x_{t\mathrm{Bu}}$ range. The values confirm that Me docking is strongly preferred for all systems. This result is completely robust with respect to the four theoretical levels, even allowing for possible ZPVE errors of ±0.2
kJ${\mathrm{mol}}^{-1}$ and for residual errors in the theoretical cross section ratio (see Appendix A for more details).

One should emphasize that the predicted energy imbalance between the two docking isomers is always below 8% (Appendix A), so rather small on an absolute scale. Our experiment is thus rather sensitive in detecting errors in this imbalance, making it suitable for benchmarking studies [10].

Figure 6a plots the (semi-)experimental fraction of *t*Bu docking (Table 1 and Appendix A) against the energy difference prediction for the four combinations of functional with basis set. The grey areas indicate qualitative inconsistencies between theoretical prediction and experiment, within the assumptions of uniform anharmonicity and accurate cross section ratio. If Me docking is energetically favorable, *t*Bu should not dominate the expansion and vice versa. Asymmetrical error bars are obtained by taking the mean value for $I_{\mathrm{Me}}/I_{t\mathrm{Bu}}$ as the data point and using the Monte Carlo determined range (see Table 1 and Appendix A) as the boundaries.

At first sight, experiment and DFT theory (Figure 6a) are consistent with each other and different DFT levels cannot be discriminated against each other. Even the obvious outliers for CpOH can be accommodated in the allowed (white) area. However, two closer looks at the data reveal deficiencies.

To bring the different theory levels closer together, one can plot the experimental *t*Bu docking abundance gain $\Delta x_{t\mathrm{Bu}}$ against the theoretical *t*Bu docking energy gain $\Delta \Delta E_{t\mathrm{Bu}}^{0}$, when switching from MeOH to a heavier alcohol (Appendix A). One would expect that any energy gain leads to a docking abundance gain, but all DFT methods predict a higher energy gain for CpOH and experiment finds a higher docking abundance gain for *t*BuOH. Clearly, the DFT description is somewhat imbalanced for either CpOH or *t*BuOH or for both.

Another way of analyzing the deficiency is to calculate an effective conformational temperature $T_{c}$ for each DFT method and pair of isomers from the experimental band integral ratio and the computed IR band strength ratio [10]. Based on the (semi-)experimental concentration ratios $\frac{c_{\mathrm{Me}}}{c_{t\mathrm{Bu}}}$ listed in Table 1 and the computed energy differences $\Delta E_{\mathrm{Me}-t\mathrm{Bu}}^{0}$ in Figure 6, this can be obtained as
$$T_{c}\approx -\frac{\Delta E_{\mathrm{Me}-t\mathrm{Bu}}^{0}}{R\ln\frac{c_{\mathrm{Me}}}{c_{t\mathrm{Bu}}}}$$
with the universal gas constant *R*, if there are no symmetry differences between the docking isomers and the rovibrational partition functions are sufficiently similar due to supersonic jet cooling. $T_{c}$ should roughly fall in the range of 30 to 150 K [11,15]. This is the case for almost all 12 combinations of system and method, within the respective error bar (Appendix A). Only TPSS for *t*BuOH–Pin gives higher $T_{c}$ values and B3LYP for CpOH–Pin is borderline on the low end. The former could be due to a higher interconversion barrier but the latter is likely due to an overestimated stability of the *t*Bu docking side.

These inconsistencies call for a check with wavefunction theory, which is presented in the next section.

### 2.3. DLPNO-CCSD(T) Check

For the large complexes of interest in this work, harmonic frequency analysis and thus zero-point energy calculation is not very practical beyond DFT level. However, single point energies at DLPNO-CCSD(T) level [25] were calculated at the minima obtained for the various DFT methods (with the setting tightPNO, basis sets aug-cc-pVQZ and aug-cc-pVQZ/C, see Appendix A). They offer several benefits.

First, they allow judging which of the DFT methods is likely closer to the true minimum by looking at the absolute CCSD(T) energies [9]. In all cases, B3LYP outperforms TPSS but in most B3LYP cases the smaller basis set gives a slightly lower energy. This may be taken as a weak indication that the B3LYP structures are closer to reality, but there could be some compensation between intra- and intermolecular degrees of freedom.

Second, one can replace the DFT electronic energy difference between isomers by the corresponding DLPNO-CCSD(T) difference and keep the structural and ZPVE contributions from the DFT level. This generates a variant of Figure 6a, in which the data points for all larger alcohols now fall close to or into the lower-right grey and thus unphysical region, where major *t*Bu docking is expected but Me docking is predominantly observed (see Figure 6b). Only MeOH stays in the physically meaningful range. The mere fact that DLPNO-CCSD(T) correction leads to such large energy difference changes casts doubt on the quality of the DFT (in particular TPSS) structures. Note that all 12 corrections (see Appendix A) promote *t*Bu docking, so the DFT error is highly systematic. For B3LYP, the corrections stay below 1 kJ${\mathrm{mol}}^{-1}$, for TPSS they always exceed 1 kJ${\mathrm{mol}}^{-1}$. Because experiment is consistent with a preference for Me docking in all cases, this likely means that the DFT structures for Me docking are relatively far from the best ones, in particular for TPSS. As the CCSD(T) corrections are quite uniform for all three alcohol–pinacolone complexes, it is plausible that the deficiency does not reside so much in the dispersion correction but rather in the functional and its description of differences in hydrogen bonding to the acceptor C=O group.

A third application of DLPNO-CCSD(T) is to provide dispersion contributions to the interaction energy in the LED scheme [16,17] (Appendix A). This is a refined way of obtaining such (strictly speaking non-observable) dispersion energies, which is conceptually better than simply evaluating the size of the D3 correction in the complex (Appendix A). In the present case, the numbers obtained for both methods are quite similar, but this cannot be generalized except perhaps for large distances, where London dispersion is best defined and LED [16], SAPT [18] or empirical dispersion correction [26] should become comparable as leading corrections to long range electrostatic and inductive interactions. Dispersion always favors *t*Bu docking, by 1.5 to 3.1 kJ${\mathrm{mol}}^{-1}$ in the LED scheme (1.6 to 2.8 kJ${\mathrm{mol}}^{-1}$ for D3 corrections). The dependence on the size of the alcohol is quite modest, but CpOH tends to show the largest gains, at least for B3LYP.

Returning to the conformational freezing temperature analysis, now with DLPNO-CCSD(T)-corrected values (Appendix A), only MeOH docking yields reasonable $T_{c}$ values (larger than 30 K). For *t*BuOH docking, B3LYP predicts borderline $T_{c}$ values and TPSS predictions are far too low. For CpOH, none of the CCSD(T)-corrected DFT results give physical $T_{c}$ values.

In summary, the DLPNO analysis shows that dispersion-corrected TPSS docking structures are imbalanced, more so than B3LYP structures. It confirms that beyond MeOH, the best isomer energy predictions are inconsistent with experiment or at best borderline (for B3LYP and *t*BuOH docking). Compared to acetophenone [11], Pin is seen to be a more critical test ketone. As it is purely aliphatic, there is likely some error compensation in the apparently more successful mixed aliphatic-aromatic acetophenone case [11].

## 3. Materials and Methods

The spectroscopic data were obtained by probing pulsed supersonic slit jet expansions of Pin+alcohol-seeded helium gas with a synchronized FTIR spectrometer. Specifically, helium (Linde 99.996%) is led through a temperature-controlled gas-flow system, where it passes separate gas saturators filled with the analytes pinacolone (Alfa Aesar > 97%) and alcohol (methanol (Roth ≥ 99.9%), *tert*-butyl alcohol (Roth ≥ 99.9%) or cyclopentanol (Fluka Chemicals > 99.9%)). The gas mixture is filled into a 67 L reservoir at a pressure of 0.75 bar and pulsed through six magnetic valves into a pre-expansion chamber which is terminated by a 600 mm long and 0.2 mm wide slit nozzle. During about 0.2 s, the gas flows through this slit into a vacuum chamber connected to a buffer volume ( 23 $m^{3}$), which is continuously evacuated by a series of pumps with a power of 500 to 2500 $m^{3}$$h^{-1}$. The expansion is crossed by a modulated and softly focused IR beam from a Bruker IFS 66v/S FTIR spectrometer with a 150 W tungsten filament, CaF${}_{2}$ optics and a liquid nitrogen cooled InSb detector. The scans are obtained with a resolution of 2 ${\mathrm{cm}}^{-1}$ and are synchronized with the gas pulse. The shown spectra are averaged over 300–425 gas pulses. More details on the experimental setup can be found elsewhere [27]. No evidence was found that more than two structural isomers of the studied 1:1 complexes are formed during the experiment.

To determine the band integral ratios $I_{\mathrm{Me}}/I_{t\mathrm{Bu}}$, an automated statistical evaluation was used, where the main entering parameters include the band positions and band width, which is statistically varied (chosen at (3.0±0.5) ${\mathrm{cm}}^{-1}$)[24]. The program adds synthetic noise to the spectra, providing statistical error bars for $I_{\mathrm{Me}}/I_{t\mathrm{Bu}}$. The resulting 95% confidence interval was used for further data processing.

DFT calculations were used for assignment purposes and to trigger future benchmarking of their ability to describe the combination of hydrogen bonding and distant London dispersion interactions. Therefore, they were limited to two functionals and two basis sets, but others are invited to find more powerful density functionals for this challenge. The initial structural search (manual and using Crest [28]) was carried out at B3LYP-D3/def2-TZVP level [29,30,31,32]. Reoptimization was carried out with a def2-QZVP basis set [32] and with the meta-GGA functional TPSS-D3 [33] using the same def2-TZVP and def2-QZVP basis sets. Three body-inclusive D3 dispersion correction [26] with Becke–Johnson damping [34,35,36,37] was always applied. Single point energies were obtained using DLPNO-CCSD(T) [25,38,39] at the DFT-optimized structures. For all these calculations, ORCA version 4.2.1 [40] was used. Further information on computational details can be found in the Appendix A). Thermal corrections to the isomer equilibrium were neglected due to the low and mode-dependent temperatures in a jet expansion, with rotational temperatures expected to be on the order 10 K. Vibrational temperatures are on the order of 100 K and conformational temperatures, which depend on the barrier between isomers, are discussed in the main text [10]. The harmonic treatment of the ZPVE is expected to be more than sufficient for this kind of systems and for the achievable accuracy, due to the near-equivalence of the two lone electron pairs [11]. A transition state search for one system was carried out with Woelfling (Turbomole [41,42]) and followed by an optimization with ORCA version 4.2.1 [40].

## 4. Conclusions

Three alcohols of increasing size were combined with pinacolone to determine the hydrogen bonding preference to either the methyl- or the *tert*-butyl-facing lone electron pair of the keto group. As generally predicted for almost two dozen alcohols by dispersion-corrected B3LYP calculations, the methyl side is preferred for methanol, *tert*-butyl alcohol and cyclopentanol. This was qualitatively confirmed by infrared spectroscopy of supersonic jet expansions in combination with approximate IR absorption cross sections. Quantitatively, the DFT predictive power in terms of the spectral splitting decreases with increasing alcohol size. In addition, the observed spectral abundance does not correlate systematically with the predicted energy difference. DLPNO-CCSD(T) energy calculations indicate that B3LYP provides a somewhat better description of the combined hydrogen bond and London dispersion interaction than TPSS. However, in combination with the experiment, they suggest that docking on the methyl side is systematically underrated by both density functionals on the 1 kJ${\mathrm{mol}}^{-1}$ scale. This only amounts to about 3% of the total binding energy but is quite significant on the relative energy scale of competitive ketone docking.

Intermolecular energy balances are thus shown to be powerful benchmarking tools to assess the ability of DFT methods to describe hydrogen bonding in competition with London dispersion. The ketone balance variety is particularly useful, as it involves systematically compensating zero-point-energy contributions and therefore allows judging electronic structure predictions in a rather direct way. For acetophenone, only a slight deficiency of the B3LYP functional could be identified [11]. For pinacolone, none of the investigated functionals comes close to describing the spectral splitting and the energetics of the docking isomerism for all three alcohols, but D3-corrected B3LYP performs satisfactorily for methanol docking and borderline for *tert*-butyl alcohol. The qualitative failure of theory to describe the experimentally observed cyclopentanol docking invites studies of related complexes, such as cyclohexanol–pinacolone and cyclopentanol–acetophenone. Larger modifications involve the use of phenol [43] and the switch from the OH chromophore to NH stretching as a probe of the conformational preference.

The goal is to find a density functional which systematically reproduces the harmonic wavenumber splitting between docking isomers within better than about 10 ${\mathrm{cm}}^{-1}$ and which provides a conformational temperature of the correct sign between about 30 and 150 K across a large number of isomeric complexes with low interconversion barrier. Furthermore, DLPNO-CCSD(T) correction should not change the energy difference between the isomers by more than about 0.5
kJ${\mathrm{mol}}^{-1}$, thus indicating a sufficiently balanced structural description. The best-performing B3LYP-D3/def2-QZVP approach in the present study only fulfills about half of these criteria for the three systems and the corresponding TPSS-D3 calculation even fewer than a quarter. Considering that some of these matches will be fortuitous, this is clearly not a satisfactory state, calling for further experimental and theoretical investigations.

## Figures and Tables

**Figure 1 molecules-25-05095-f001:**
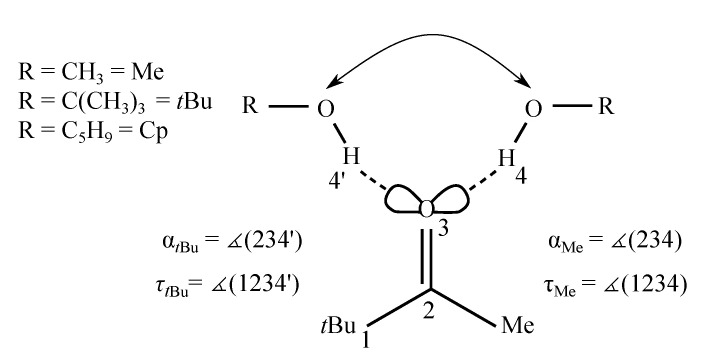
Schematic representation of the two possible docking sides 4 and 4${}^{\prime }$ in a pinacolone molecule (*t*Bu and Me) with different alcohols (R-OH, with the abbreviations Me for methyl, *t*Bu for *tert*-butyl and Cp for cyclopentyl as R).

**Figure 2 molecules-25-05095-f002:**
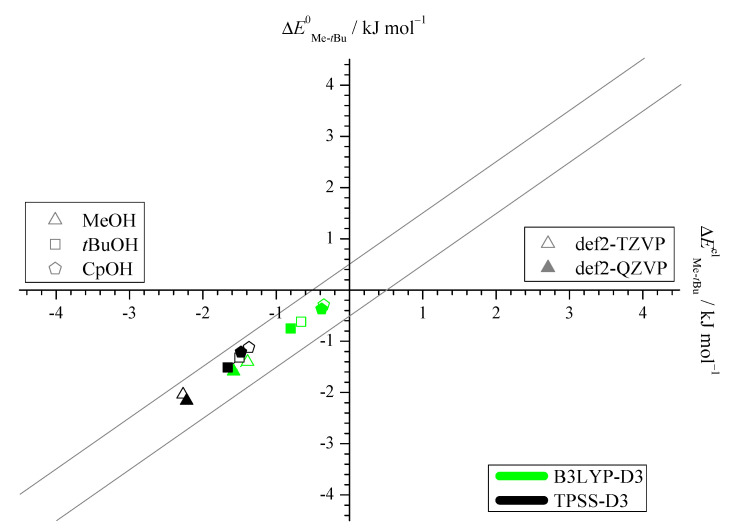
Harmonically zero-point corrected energy differences $\Delta E_{\mathrm{Me}-t\mathrm{Bu}}^{0}$ plotted against the electronic energy differences $\Delta E_{\mathrm{Me}-t\mathrm{Bu}}^{\mathrm{el}}$ referenced to the *t*Bu side, computed at B3LYP-D3 (green) and TPSS-D3 (black) level, each with a def2-TZVP (empty symbols) and def2-QZVP (filled symbols) basis set. The electronic energy differences are seen to be a good approximation to experimentally relevant ZPVE-inclusive differences and the methyl docking side is systematically preferred (see also Appendix A).

**Figure 3 molecules-25-05095-f003:**
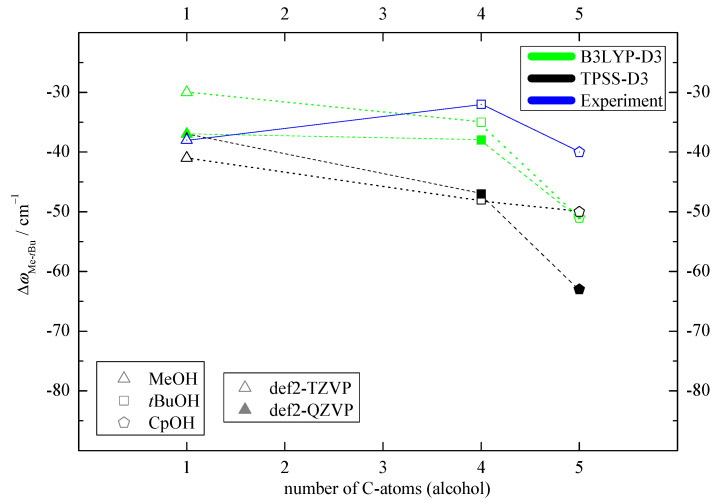
Computed OH wavenumber difference between the two docking sides $\Delta \omega_{\mathrm{Me}-t\mathrm{Bu}}$ relative to the number of C-atoms of the corresponding alcohol. This shows that the employed computational methods predict the same spectral trends for MeOH and *t*BuOH, indicated by dashed lines. For CpOH a somewhat larger discrepancy can be observed, with the smaller basis set TPSS result differing most from the experimental trend (blue) (see Appendix A for details).

**Figure 4 molecules-25-05095-f004:**
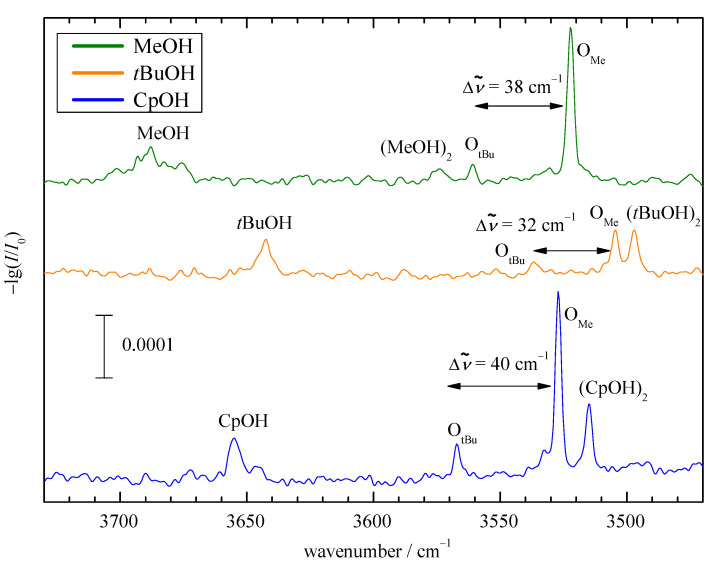
FTIR jet OH stretching spectra of Pin with the three alcohols. The 1:1 complexes are marked with O, indexed by the assigned docking preference. Both docking sides are observed. Pin is only a stronger OH shifting partner than the alcohol itself for MeOH.

**Figure 5 molecules-25-05095-f005:**
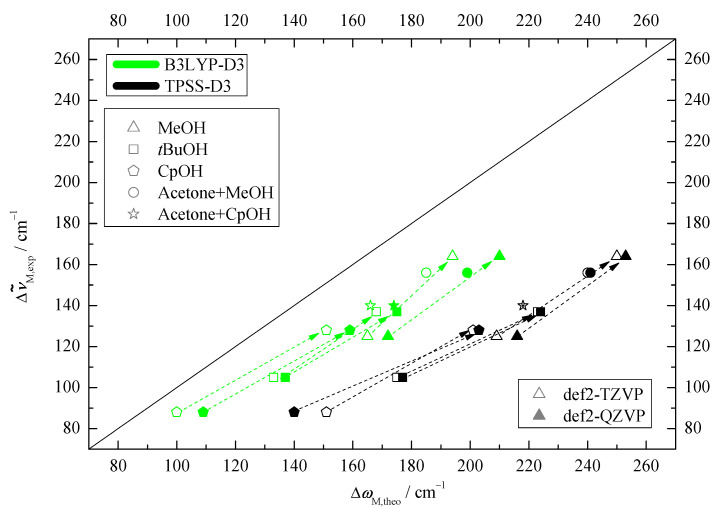
Experimental (anharmonic) downshift of the 1:1 complexes $\Delta {\tilde{\nu }}_{M,\exp}$ plotted against the harmonically computed downshifts $\Delta \omega_{M,\mathrm{theo}}$ for four computational variants. The harmonic DFT overestimation and the trend between docking sides (dashed arrows from *t*Bu to Me docking) are uniform.

**Figure 6 molecules-25-05095-f006:**
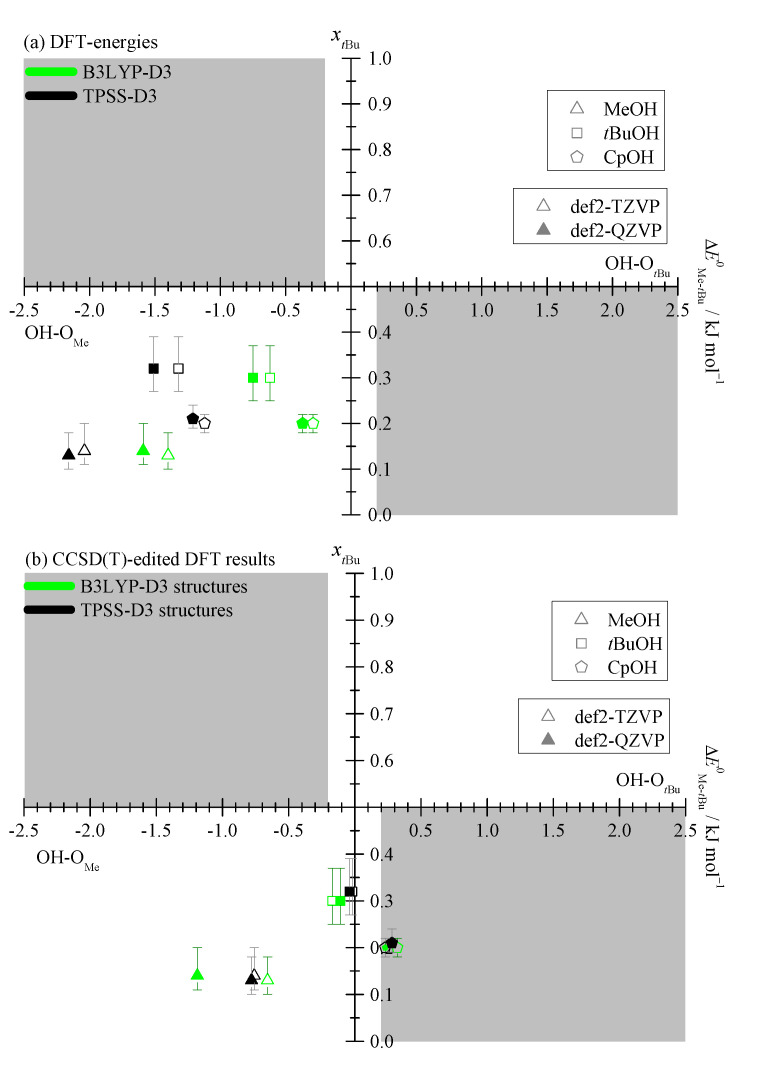
Experimental *t*Bu docking fraction $x_{t\mathrm{Bu}}=c_{t\mathrm{Bu}}/\left(c_{t\mathrm{Bu}}+c_{\mathrm{Me}}\right)$ (based on 95% confidence intervals and the mean value for the ratio $I_{\mathrm{Me}}/I_{t\mathrm{Bu}}$ from Table 1 and Appendix A) plotted against the computed ZPVE corrected energy differences $E_{\mathrm{Me}-t\mathrm{Bu}}^{0}$. Grey areas indicate inconsistency between experiment and theory, when allowing for an estimated anharmonic ZPVE error of ±0.2
kJ${\mathrm{mol}}^{-1}$ and assuming correct cross section ratios form the respective theoretical model. (**a**) DFT energies, where all models predict the correct qualitative docking preference, but the correlation of energy and abundance is non-uniform. (**b**) As in (**a**), but with the electronic energy being replaced by the corresponding DLPNO-CCSD(T) value (see Section 2.3).

**Table 1 molecules-25-05095-t001:** Experimental integrated intensity ratios $I_{\mathrm{Me}}/I_{t\mathrm{Bu}}$, B3LYP-D3(BJ,abc)/def2-TZVP cross-section derived docking ratios $c_{\mathrm{Me}}/c_{t\mathrm{Bu}}$ and resulting experimental fractions $x_{t\mathrm{Bu}}$ for *t*Bu docking. The given ranges represent 95% confidence for $I_{\mathrm{Me}}/I_{t\mathrm{Bu}}$ using an automated statistical evaluation [24] and are carried on to $c_{\mathrm{Me}}/c_{t\mathrm{Bu}}$ and $x_{t\mathrm{Bu}}$ without including a theoretical cross section ratio uncertainty.

Donor	$\frac{I_{\mathrm{Me}}}{I_{t\mathrm{Bu}}}$	$\frac{c_{\mathrm{Me}}}{c_{t\mathrm{Bu}}}$	$x_{t\mathrm{Bu}}$
MeOH	5.9–11.8	4.5–9.0	0.10–0.18
*t*BuOH	2.4–4.2	1.7–3.0	0.25–0.37
CpOH	5.5–7.3	3.4–4.4	0.18–0.23

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
