# Peer review of "Pinacolone-Alcohol Gas-Phase Solvation Balances as Experimental Dispersion Benchmarks"

_molecules, 2020, doi:10.3390/molecules25215095_

Round 1
Reviewer 1 Report
This is an interesting and generally convincing paper about Pinacolone-alcohol gas-phase solvation balances equilibria, studied experimentally by infrared spectroscopy in supersonic jet, as well as, by carefully performed calculations with two DFT methods (B3LYP-D3 and the meta-GGA functional TPSS-D3) using the def2-TZVP and def2-QZVP basis set and by single point energy calculation using DLPNO-CCSD(T) on DFT geometries. The work appears to have been carried out with care and skill, therefore I recommend publication in its current form.
A minor point: the authors could also use the APFD functional, which also takes into account the dispersion forces.
Reviewer 2 Report
This is excellent work, using a combination of detailed experiments and theoretical simulation to probe the binding of alcohols to an asymmetric ketone. The subtle balance of hydrogen bonding and dispersion interactions make this a tricky task for even the most modern DFT methods: it is shown that two such methods perform reasonably well but with clear and important shortcomings.
This is only possible due to the careful experimental design: I am not an expert in that field, so hope another reviewer can assess the details of this. My only queries there are
- how reliable "peak-picking" is in all cases, for example the one marked O_tBu in Figure 4 does not seem significant over background noise in two spectra.
- How much does interpretation depend on theoretical absorption cross section? If this is even slightly wrong, how much would that affect derived data?
Calculations are performed to a very high level using a range of suitable methods. It must be noted that the differences sought are right at the limit of the accuracy one can demand of such methods. Most people would be happy to be within 1 kcal/mol, so differences of around 1 kJ/mol are very challenging indeed! Nevertheless some success is observed.
I only have minor suggestions here:
If it were me, I would be tempted to only retain larger basis set data in main text, and move TZVP data to Supplementary. Basis set dependence is important, but doubling the number of points in each figure can get confusing.
I would like to see cartesian coordinates added to supplementary.
Reviewer 3 Report
In the presented article, the authors present theoretical and spectroscopic studies for pinacolone heterodimers with three alcohols (methanol, tert-butanol, and cyclopentanol). The research is described in detail, but not very easy to read. I have no substantive objections, but it seems to me that the paper could be shortened. I don't think the Cp ring can be called flat (line 266-267).
The paper would be more valuable if the authors could find a computational method that could reproduce the experimental results at satisfactory level.
Reviewer 4 Report
The manuscript entitled “Pinacolone-alcohol gas-phase solvation balances as experimental dispersion benchmarks” done by M.A. Suhm & colab. presents a detailed theoretical and experimental investigation about the importance of the correct description of the London-type dispersion forces for obtaining accurate gas-phase solvation balances in pinacolone-alcohol mixture. The research topic explored in the manuscript is of great importance since it useful for estimating the level of accuracy of some widely used DFT-D3 frameworks. The manuscript is well written and presents an interesting methodological discussion and analysis. The aims are clearly formulated and the discussion of the results follows the logical sequence of these aims.
I would like to add only two short comments:
- As claimed by Bistoni et al (see Ref.-s 16 and 17) direct comparison for the amount of the dispersion corrections obtained in the DLPNO-CCSD(T)-LED and in the semi-empirical Grimme’s D3 frameworks cannot be done. Of course, the Authors are aware of this, but a short warning text about it should be included for the readers.
- Frequency shifts induced separately by the H-bond interaction, anharmonicity or BSSE effects on the harmonic vibrational frequency was studied in detail in Theoretical Chemistry Accounts 125 (3-6), 253-268. It would be important to include in the manuscript a brief remark about the role of these effects.
The manuscript can be considered for publishing in Molecules journal after a minor revision.
